# Turning Microbial AhR Agonists into Therapeutic Agents via Drug Delivery Systems

**DOI:** 10.3390/pharmaceutics15020506

**Published:** 2023-02-03

**Authors:** Matteo Puccetti, Marilena Pariano, Paulina Wojtylo, Aurélie Schoubben, Stefano Giovagnoli, Maurizio Ricci

**Affiliations:** 1Department of Pharmaceutical Sciences, University of Perugia, 06123 Perugia, Italy; 2Department of Medicine, University of Perugia, 06132 Perugia, Italy

**Keywords:** drug delivery, postbiotics, aryl hydrocarbon receptor, indole-3-aldehyde, anakinra

## Abstract

Developing therapeutics for inflammatory diseases is challenging due to physiological mucosal barriers, systemic side effects, and the local microbiota. In the search for novel methods to overcome some of these problems, drug delivery systems that improve tissue-targeted drug delivery and modulate the microbiota are highly desirable. Microbial metabolites are known to regulate immune responses, an observation that has resulted in important conceptual advances in areas such as metabolite pharmacology and metabolite therapeutics. Indeed, the doctrine of “one molecule, one target, one disease” that has dominated the pharmaceutical industry in the 20th century is being replaced by developing therapeutics which simultaneously manipulate multiple targets through novel formulation approaches, including the multitarget-directed ligands. Thus, metabolites may not only represent biomarkers for disease development, but also, being causally linked to human diseases, an unexploited source of therapeutics. We have shown the successful exploitation of this approach: by deciphering how signaling molecules, such as the microbial metabolite, indole-3-aldehyde, and the repurposed drug anakinra, interact with the aryl hydrocarbon receptor may pave the way for novel therapeutics in inflammatory human diseases, for the realization of which drug delivery platforms are instrumental.

## 1. Introduction

Changes in gastrointestinal microbiome composition and function can adversely impact host physiology and are associated with local and systemic inflammatory diseases, cancer, metabolic disorders, and opportunistic and recurrent pathogen infections. It is estimated that noncommunicable illnesses contribute to 70% of worldwide mortality. Due to the presence of biosynthetic or metabolic gene clusters encoding the biosynthetic machinery to produce primary and specialized metabolites [1], microbes are highly metabolically active and produce trans-kingdom signaling molecules that interact with competing microorganisms and the host [2]. The fermentation of undigested protein in the large intestine results in the formation of branched-chain amino acids, biogenic amines, short-chain fatty acids, ammonia, phenolic and indolic compounds, hydrogen sulfide, and nitric oxide [3]. All these metabolites play important pathophysiological roles [4] such that their identification along with the study of their effector pathways can lead to the discovery of incredibly innovative diagnostics and therapeutics. According to the Virtual Metabolic Human database, 5607 metabolites involved in intestinal microbiota metabolism in humans, collectively termed the metabolome, have already been identified [5].

It is now clear that metabolic host–microbe interactions constitute a central hub that integrates immunological, neuronal, and endocrinal responses to environmental and mental stimuli. Thus, the study of the chemistry behind the microbe–microbe and host–microbe metabolic interactions may provide a useful platform for discovering multi-level acting compounds [6]. Beneficial metabolites are promising in this regard. Indeed, due to the complexity of multifactorial diseases, drug intervention based on single-target drugs with high affinity, high selectivity, and strong potency may not fit well and does not always exhibit satisfactory efficacy with the network-based, inter-balanced regulation mode of the smart biological system [7]. This has been a major bottleneck in the translation of potent single-target candidates, which inherently possess excellent potential but fail to demonstrate significant clinical impact due to disease mechanisms, which are in fact complex subnetworks within the interactome [8,9]. Moreover, disease definitions are mostly symptom-based rather than mechanism-based, and the therapeutics are likewise. The holistic picture of multiple metabolites engaging with multiple targets, i.e., diverse metabolites interacting with a shared target and/or a single metabolite binding to diverse targets, represents a unique feature of these signal transduction systems that demands a multitarget-directed approach [10] that, according to the paradigm of “network pharmacology” [8], requires the creation of a single molecule with multi-target properties.

In this review, we highlight how drug delivery technologies are essential in the development of successful therapeutic metabolites, referred to as postbiotics, that could improve patient health by enhancing delivery to target sites, minimizing off-target accumulation and facilitating patient compliance. Specifically, we discuss how the target delivery of the microbial metabolite indole-3-aldehyde (3-IAld) may turn a multifunctional compound into an effective therapeutic capable of orchestrating host pathophysiology and microbial symbiosis. More generally, our study may have important translational implications from providing evidence of how to overcome the hurdles of targeting promiscuous receptors to the exploitation of the endogenous metabolome as a pharmacology baseline for drug discovery.

## 2. Metabolite-Based Therapeutics: The Postbiotic Concept

The metabolome is contributed by diet, the host, and its microbiota. Under homeostatic conditions, the intestinal microbiota produce, modify, and degrade metabolites which serve as an effective means of communication in host–microbe interactions and profoundly affect human health. Metabolites are bioactive, and their functional activity includes the modulation of signaling and metabolic pathways that contribute to mucosal immune homeostasis [10,11]. Despite the fact that the microbiome is estimated to account for more than half of all fecal and urinary metabolites, only about a dozen metabolites currently have a well-characterized effect on the host; these include target cells, receptors, signaling pathways, and physiological outcomes [12].

Quantitative or qualitative variation in metabolites is perceived by the host as an alteration of its microbial community, potentially an early indication of dysbiosis. Dysbiosis leads to subsequent alterations in metabolite composition, which has been shown to have direct consequences on host health in the context of multiple diseases. Thus, metabolite-based therapeutics, or “postbiotics”, may work as replacement therapy that is potentially effective against dysbiosis and its inflammatory consequences [11]. In addition, metabolites are also sensed by microbes themselves, meaning that they affect bacterial functions and activities. In this regard, by acting on indole-producing and non-indole-producing bacteria, without the requirement for specific receptor binding, indole is paradigmatic in that it affects, among others, spore or biofilm formation, drug resistance, and virulence [13].

Metabolites have a clear chemical structure and a long shelf life, occur naturally in a broad range of concentrations, are functionally pleiotropic, are easy to administer, are suitable for different routes of administration, show tissue bioavailability, are present at most body sites, are generally stable in the blood, and are therefore amenable for scalable modulation of their concentration. Because of these properties, they are attractive and uniquely suited for therapeutic strategies. However, although highly promising, biopharmaceutical approaches for their in vivo delivery are critically needed to turn beneficial pleiotropic activity into effective therapeutics for human illnesses. This demands for drug delivery systems (DSS) that pivotally contribute to the development of an ideal postbiotic capable of fostering the holistic balance for the homeostatic control of inflammation at the mucosal and tissue levels.

## 3. Tissue-Specific Drug Delivery

The field of drug delivery systems, including routes of delivery and delivery vehicles, has advanced dramatically in the past few decades, thanks to the increased understanding of the physiological barriers for efficient drug delivery and the development of several new modes of drug delivery that have entered clinical practice. These include lipid nanoparticles (essential for effective delivery of mRNA), the microneedle patch (for sustained delivery to low-income people or people with mental-health conditions), the robotic pill (for oral drug delivery of complex drugs), and cellular delivery via red blood cells or bacteria, to name a few [14]. However, despite improving the drug’s efficacy, side effects may still occur because the unwanted drugs interact with healthy organs or tissues. Thus, targeted drug delivery approaches are of great interest in the pharmaceutical sciences, as they enable the concentrated delivery of a drug compound to its desired target—increasing efficacy and reducing off-target effects as compared to conventional formulations [14]. DDS have enabled the development of many pharmaceutical products that improve patient health by enhancing the delivery of a therapeutic to its target site, minimizing off-target accumulation and facilitating patient compliance [15]. An emerging concept is that a drug and its delivery vehicle must work as a team to accomplish a therapeutic objective. It is often said that the aim of some advanced delivery systems is a matter of getting the right drug to the right part of the body at the right time. The breakthroughs in microtechnology and especially nanotechnology, and the availability of a number of platforms in the nanometer size range have brought about revolutionary innovations in biological research techniques and medical practice. Targeted delivery systems based on nanoparticles, for example, and various carrier systems have been described for targeted delivery or/and release of postbiotics in various in vivo models [16]. However, significant pharmaceutical advances are needed for the oral and lung delivery of postbiotics. Despite the challenges of each route—such as the different pH, enzymatic activity, and mechanical forces and the presence of a mucus layer and local defense mechanisms—targeted delivery based on dry powder inhalation therapy has been shown to be an effective method for treating respiratory and non-respiratory diseases [17,18]. We have resorted to dry powder formulations for the organ-targeted delivery of postbiotics and for drug repurposing. In doing so, we obtained a proof-of-principle demonstration of how biopharmaceutical technologies pivotally contribute to optimize AhR agonists into therapeutic agents via DDS.

## 4. The Ubiquitous AhR Signaling Pathway

The aryl hydrocarbon receptor (AhR) is a ligand-activated transcription factor that acts as a heterodimeric transcriptional regulator [19]. Animal and human data indicate that the AhR is involved in various signaling pathways critical to normal cell homeostasis, which covers multiple aspects of physiology, such as cell proliferation and differentiation, gene regulation, cell motility and migration, inflammation, and host–microbiota interplay [20,21]. Signal molecules function as ligands for AhR, and activated AhR forms heterodimers at promoter recognition sequences of the target genes. The AhR/AhR nuclear translocator complex may then require coactivators (including members of other families of transcription factors) in order to initiate transcription and to unwind histone-bound DNA for exposing additional promoter recognition sites via their histone acetyltransferase function. Within this scenario, three major factors appear to contribute to the outcome of gene transcriptional regulation by AhR: the nature of the ligand, the local tissue microenvironment, and the presence of coactivators in the cell [19,20]

The gut and lung are very responsive to AhR stimuli, suggesting that AhR ligands—whether natural or synthetic—are promising drugs as regulators of inflammatory pathology at mucosal surfaces. Xenobiotics including dietary phytochemicals, host and microbial metabolites, and ubiquitous environmental pollutants may have shaped this system in mucosal epithelia during millions of years of evolution. In the skin, for instance, AhR ligation controls oxidation/antioxidation, epidermal barrier function, photo-induced response, melanogenesis, and innate immunity [22]. In overinflammatory systemic responses induced by infection and other noxae, host tryptophan (Trp) catabolic enzymes produce l-kynurenine that suppresses inflammation via AhR [23]. The important role played by host kynurenine via signal onto AhR is well established [24]. However, microbial Trp metabolites are also sensed by AhR and other host xenobiotic receptors [25]. A number of studies have highlighted the ability of AhR to respond to indoles and indolyl metabolites, of both microbial and dietary origin, thus positioning AhR as a candidate indole receptor of biological relevance [26].

## 5. The Tryptophan-Indole Microbial Pathway

Trp is an essential amino acid supplied by dietary protein whose degradation occurs throughout the small intestine [27,28]. The gastrointestinal tract harbors numerous species with the capacity to synthesize indole and indole-containing compounds. The bacterial synthesis of indole compounds occurs via different metabolic pathways mainly involving the activity of tryptophanase, aromatic amino acid, aminotransferases, and tryptophan deaminases generating indole and a number of indole derivatives [29]. Indoles exert significant biological effects, including immune modulation, epithelial barrier regulation, and pathogen colonization, through which they may contribute to the etiology of a variety of human diseases, including dysbiosis [30]. More recently, an impaired microbial Trp catabolism was associated with critical COVID-19 patients [31]. These observations suggest that the exploration of indoles to support beneficial bacterial/host symbiosis and cooperation may lead to the rational design of effective clinical interventions in human chronic inflammatory conditions in which the epithelial barrier disruption and microbial dysbiosis are causally linked [32]. From an evolutionary perspective, many of the endogenous Trp metabolites exhibit greater activation potential for the human AhR when compared to the rodent AhR, an observation highlighting the importance of the AhR in human barrier tissues [26]. Despite differences in the reported concentrations, indole and indole derivatives have been detected in human fecal samples (at both the mM and µM ranges), serum (within the µM ranges), and urine (within the µM ranges) [27], thus suggesting that the indole–AhR pathways are druggable targets in humans. However, due to the fact that AhR activation is both context- and ligand-dependent, this demands suitable—most likely, innovative in their inception—biopharmaceutical formulations that enable site-specific (“targeted”) delivery, so as to allow for therapeutic efficacy in the absence of off-target effects, including systemic deregulation of immune homeostasis.

## 6. Turning Indoles into Therapeutics via Targeted Delivery Technologies

Within Lactobacilli, *L. reuteri* was found to convert Trp to indole-3-aldehyde (3-IAld) via the aromatic amino acid aminotransferase [33]. Mice exposed to a tryptophan-enriched diet expand *L. reuteri* in the gut that produce 3-IAld; this promotes AhR-dependent transcription of the IL-22–encoding gene by host innate lymphoid cells and thus prevents microbial infections and local inflammation [33]. 3-IAld, like indoles, moonlights as a metabolite and signaling molecule and is increasingly being associated with the regulation of wide-ranging physiological processes [34]. Being rapidly metabolized upon administration [35], 3-IAld may potentially act as a rapidly metabolized AhR ligand within the range of concentrations found in murine and human gut, despite not being among the most active endogenous ligands of murine and human AhR [36,37]. In murine models, endogenous or administered 3-IAld was associated with resistance to infections, ionizing radiation, inflammaging, and gut, cerebral, and liver inflammation via a plethora of mechanisms, including the production of IL-22 and type I interferons, the modulation of IL-10 and IL-10R expression, NF-kB and TLR7 activity, the promotion of epithelial barrier function, and cross-talk with enteric neurons and the microbiota [33,34,38,39,40,41,42]. In weaned piglets, dietary supplementation of 3-IAld significantly increased the jejunal, ileal, and colonic indexes and modulated the intestinal flora [43]. Of interest, an alteration in the microbial composition may change the levels of 3-IAld in human diseases. For instance, 3-IAld production was reduced in patients with atopic dermatitis [42,44], inflammatory bowel disease [45], and celiac disease [46] and in hematologic patients with fungal pneumonia [47]. These findings suggest that replacement therapy with 3-IAld may be of benefit in human inflammatory diseases.

Recently, resorting to targeted delivery technologies has highlighted the possible translation to a clinical application of 3-IAld. Targeted delivery of 3-IAld via dry powder inhalation or orally was an efficient strategy to restore immune and microbial homeostasis in [35,48,49] in preclinical models of lung or gut inflammation (Figure 1).

Inhaled dry powder of 3-IAld demonstrated a good pharmacological and toxicological profile upon delivery for pulmonary administration and was superior to other administration modalities (oral and intranasal) in reducing the disease score [50]. When assessed in a relevant preclinical model, the site-specific delivery of 3-IAld was an efficient strategy to control infection and restore immune homeostasis in the lungs of mice with cystic fibrosis infected with *Aspergillus fumigatus* conidia. 3-IAld exerted significant control on the infection and ensuing inflammatory response administered either before (P) or after (T) the fungal challenge, as revealed by the reduced fungal growth, lung inflammatory pathology, and neutrophils recruitment (Figure 1A). Treatments also reduced the expression of cytokines known to be involved in inflammation, epithelial dysfunction, and mucosal immune dysregulation [48]. Importantly, no signs of AhR activation were observed in distant organs. These results suggested that inhaled 3-IAld could be of therapeutic value in maintaining lung immune homeostasis in CF, at a reduced dose and without systemic activity.

3-IAld was also effective when microencapsulated on enteric microparticles (MP) for intestinal delivery. Although spray-drying is not a conventional preparation process for enteric formulations, we found it suitable in the manufacturing of gastro-resistant 3-IAld [49]. As a matter of fact, we demonstrated that 3-IAld is released from MP in the intestine where it activates AhR [35]. Pharmacokinetics studies revealed that 3-IAld was detected at high levels in the intestine within 1 h after its administration at very low levels in the serum, lung, liver, and brain, suggesting a rapid local and/or systemic transformation of the molecule, likely occurring via host and/or microbial metabolic pathways, including AhR-induced Cytochrome P450 genes. However, AhR-activated genes were detected long after the sharp dropping of 3-IAld levels and could also be detected at distant organs, a finding suggesting that 3-IAld could be metabolically converted into downstream AhR agonists. Our unpublished observation (Pariano M. and Puccetti M.) confirms that a number of 3-IAld-derived metabolites with AhR agonistic activity and biological activity in vivo are detected in different organs of 3-IAld-treated mice. Whatever the nature of the metabolite(s), we found that 3-IAld-MP was very active in a murine model of primary sclerosing cholangitis (PSC), a long-term liver disease characterized by a progressive course of cholestasis with liver inflammation and fibrosis. Given the contribution of the intestinal barrier dysfunction and the consequent bacterial/bacterial product translocation to the PSC pathogenesis [51], the ability of 3-IAld to preserve epithelial barrier integrity and maintain local immune homeostasis [35] would predict a beneficial effect in PSC. We found that 3-IAld-MP reduced liver injury and fibrosis in C57BL/6 mice fed with a 3,5-diethoxycarbonyl-1,4-dihydrocollidine (DDC)-containing diet (active phase) and subjected to a recovery time (recovery phase) as revealed by the reduced serum levels of alanine aminotransferase (ALT), alkaline phosphatase (ALP), and total bilirubin. 3-IAld-MP also reduced fibroblast activation and collagen deposition (Figure 1B) in DDC-fed mice (Figure 1B). We are inclined to believe this is also an effect mediated by 3-IAld. However, further studies are needed to define the activity of 3-IAld directly in the liver, considering that contrasting results have been obtained regarding the probiotic effect of *L. reuteri* in liver diseases. Indeed, *L. reuteri* treatment reversed the phenotype of ethanol-induced hepatitis and metabolic disorders in one study [52], but *L. reuteri* producing 3-IAld also induced autoimmune liver pathology in a condition of lack of the hematopoietic Tet methylcytosine dioxygenase 2 [53]. Whatever fine mechanisms lie behind this result, these findings suggest that the ability of 3-IAld to act upon the “gut–liver axis” occurs by restoring mucosal integrity and modulating the intestinal microbiota (see below).

On the basis of the same mechanism of activity, it was not surprising to find a protective role for 3-IAld-MP in the murine model of metabolic syndrome, a disorder characterized by a cluster of diseases associated with the reduced capacity of the microbiota to metabolize Trt into AhR ligands [54]. Accordingly, we showed that 3-IAld-MP, locally released, decreased inflammatory pathology, restored tissue architecture, and induced epithelial cell proliferation in C57BL/6 mice on a high-fed diet (Figure 1C). Concomitantly, 3-IAld-MP prevented the metabolic complications associated with the metabolic syndrome, such as hyperglycemia (Figure 1C). These results highlight the role of gut microbiota–derived metabolites, including 3-IAld, as a biomarker and as a basis for novel preventative or therapeutic interventions for metabolic disorders. Finally, 3-IAld-MP protected mice from intestinal damage by activating the AhR/IL-22-dependent pathway in a murine model of immune checkpoint inhibitor (ICI)-induced colitis as shown by the prevention of weight loss and disease activity and by the amelioration of gross and colon intestinal pathology (Figure 1D). Of interest, 3-IAld affected the composition and function of the microbiota, thus working via dual action on both the host and the microbe sides. We found that fecal microbiota transplantation from 3-IAld-treated mice protected against ICI-induced colitis with the contribution of butyrate-producing bacteria. Metabolites other than 3-IAld, including members of short chain fatty acids, were indeed measured in the sera of mice with ICI-induced colitis and treated with 3-IAld-MP; they likely contributed to the unaffected antitumor activity of ICIs in the presence of 3-IAld [41]. Indeed, 3-IAld was found to activate CD8+T cells for antitumor activity via AhR once produced at the tumor site and to fail to promote tumor growth when administered in tumor-bearing mice [55]. Thus, despite the finding that indole-producing Lactobacilli drastically increase tumor size by influencing immunity in the tumor microenvironment [55], these findings highlight the ligand-dependent activity of AhR and, at the same time, expand upon the therapeutic activity of 3-IAld. These studies provide proof-of-concept demonstration of how pharmaceutical technology helps in translating the new frontiers of microbial drug discovery into human therapeutics.

## 7. Targeted Delivery for Drug Repurposing: The Case of Anakinra

Traditional methods of drug discovery constitute a complex, costly, and risky process. Drug repurposing is increasingly becoming an attractive proposition because it involves the identification of new therapeutic purposes of existing or already marketed drugs [56]. However, drug repurposing could also be effective for generating drug candidates with new therapeutic indications based on unique formulation and/or a dosage regimen that cannot easily be achieved with the available generic versions of the drug. The case of anakinra is a teaching example of the successful feasibility of this last approach. Inflammasomes are protein complexes of the innate immune system that initiate inflammation in response to either exogenous pathogens or endogenous danger signals [57]. Activation of the inflammasome leads to the activation of caspase-1, which cleaves pro-inflammatory cytokines such as IL-1β and IL-18, providing protection against infectious pathogens but also mediating control over sterile insults. However, aberrant inflammasome signaling has been implicated in the development of cardiovascular and metabolic diseases, cancer, and neurodegenerative disorders [58]. Although the anti-IL-1β strategy has been successful in preventing pathogenic inflammatory events [59], it could be associated with a higher incidence of infections due to the systemic suppression of the immune system. We have recently obtained proof-of-concept demonstration that targeted delivery technology may help overcome this problem. Anakinra, the recombinant form of IL-1Ra, from which it differs in the presence of an extra methionine in the amino terminus and the absence of glycosylation, is currently used to treat a wide range of diseases that go beyond its approved indications for rheumatoid arthritis and cryopyrin-associated periodic syndromes [60]. Its therapeutic potential derives from its ability to prevent the production of inflammasome NLRP3-dependent IL-1β and the subsequent activation of pathogenic Th17 cells [61] (Figure 2, left side). More recently, however, we have demonstrated that anakinra promoted AhR activation in the lung through the IDO1-kynurenine pathway, independent of the IL-1R1 inhibition activity [62]. Specifically, anakinra promoted H_2_O_2_-driven autophagy through AhR that, activated through IDO1-kynurenine pathway, transcriptionally activated NADPH oxidase 4 (NOX4) and Superoxide dis-mutase (SOD)2 independent of the IL-1R1 (Figure 2, right side). By acting at the intersection of mitochondrial oxidative stress and autophagy, anakinra may have the capacity to restore conditions in which defective proteostasis leads to human disease. Based on this knowledge, we have developed an inhalable dry powder of anakinra to meet the specific needs of lung drug delivery. The new formulation, when delivered to the lungs of mice with CF, showed an improved and extended therapeutic efficacy as well as higher potency compared to the systemic delivery of equivalent doses. Peripheral side effects, such as neutropenia and inhibition of inflammatory signaling pathways in the spleen, were not observed and correlated with the lower serum levels compared to the parenteral injection. Importantly, a rough estimation of the human equivalent dose corresponding to the maximum dose employed in our study showed that this value is about half of the daily clinically prescribed systemic dose of anakinra [63]. Therefore, our results provide a proof-of-concept demonstration of the prspective potential of anakinra repurposing in CF via the pulmonary route at a reduced dosage and reduced risk of side effects.

## 8. Conclusions

Given the pervasiveness and pleiotropy of metabolite effects on human physiology, the translational potential of these molecules for “postbiotic” therapies is enormous. Recent advances in formulation and delivery strategies have facilitated the transformation of product portfolios and development pipelines by several classes of compounds. For AhR modulators, virtual screening utilizing both ligand-based and structure-based methods provided large databases of small molecules as novel AhR agonists or antagonists ready to be validated in biological assays. This demands appropriate formulation and suitable delivery technology for the tissue- and cell-specific targeting of AhR. In this regard, nanoparticles have been engineered to deliver ligands and other compounds in vivo to induce specific cell targeting of AhR [64]. We have shown the promise of drug therapy with postbiotics and repurposed drugs as AhR modulators. The success of these studies would provide great promise for AhR as a therapeutic for immunomodulation.

## Figures and Tables

**Figure 1 pharmaceutics-15-00506-f001:**
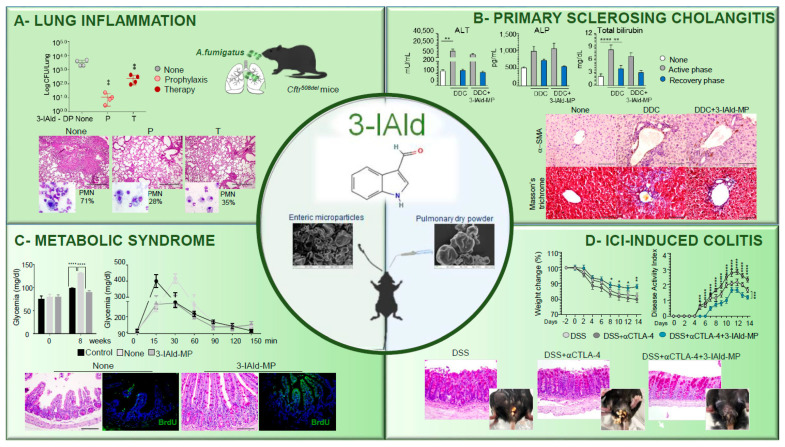
Therapeutic activity of 3-IAld in experimental models of human inflammatory diseases. (**A**) Pulmonary insufflation of 3-IAld dry powder (DP) either before (P, prophylaxis) or after (T, therapy) prevents lung inflammatory pathology in *Cftr^F508del/F508del^* mice infected with *Aspergillus fumigatus* conidia, as revealed by the fungal growth, lung histopathology, and neutrophil recruitment ([48]). Photographs were taken using a high-resolution Olympus DP71 microscope using a 10× objective. Scale bar 400 μm. Values represent the mean ± SD of four mice per group or are representative of three experiments. Naïve, uninfected mice; None, infected mice. ** *p* < 0.01, one-way ANOVA—Bonferroni’s, P or T vs. None. (**B**) 3-IAld microparticles (MP) reduces liver injury and fibrosis in C57BL/6 mice fed with a DDC-containing diet (active phase) and subjected to a recovery time (recovery phase) as revealed by serum levels of alanine aminotransferase (ALT), alkaline phosphatase (ALP), total bilirubin, fibroblast activation, and collagen deposition ([38]). Photographs were taken with a high-resolution microscope (Olympus BX51), 40× magnification (scale bars, 100 μm). ** *p* < 0.01, **** *p* < 0.0001, DDC-treated vs untreated (none) mice, and DDC vs. DDC + 3-IAld-MP. One-way ANOVA, Bonferroni post-test. (**C**) Oral administration of 3-IAld-MP decreases glycemia, promotes glucose tolerance, restores tissue architecture in the ileum, and promotes epithelial cell renewal (BrdU) in C57BL/6 mice on a high-fed diet. Control; control mice. None; high-fed diet-mice with MP alone ([35]). Photographs were taken with a high-resolution microscope (Olympus BX51), 20× magnification (scale bars, 200 μm). **** *p* < 0.001, 3-IAld-MP-treated vs MP-treated mice and None vs Control. (**D**) 3-IAld-MP protects mice from immune check-point inhibitor (ICI)-induced colitis. C57BL/6 mice were treated with DSS and anti-CTLA-4 mAb with and without 3-IAld and assessed for weight change, disease activity index, colon histopathology, and gross pathology (insets) ([41]). Photographs were taken with a high-resolution microscope (Olympus BX51), ×20 magnification (scale bars, 200 µm). wo-way analysis of variance, Bonferroni post hoc test. * *p* < 0.05, ** *p* < 0.01, *** *p* < 0.001, **** *p* < 0.0001. anti-CTLA-4 + 3-IAld- versus anti-CTLA-4-treated mice.

**Figure 2 pharmaceutics-15-00506-f002:**
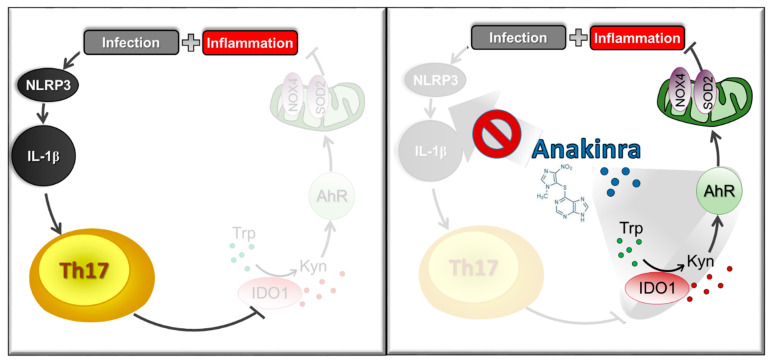
Therapeutic potential of anakinra in lung inflammation. Anakinra prevented the inflammasome NLRP3-dependent IL-1β production and the subsequent activation of pathogenic Th17 cells (**left side**) but also promoted AhR activation through the IDO1-kynurenine pathway, independent of the IL-1R1 inhibition activity. Via AhR, anakinra transcriptionally activated NADPH oxidase 4 (NOX4) and Superoxide dismutase (SOD) 2, eventually leading to autophagy-dependent control of inflammation (**right side**) (see text for the explanation).

## Data Availability

Not applicable.

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
