# Peer review of "Turning Microbial AhR Agonists into Therapeutic Agents via Drug Delivery Systems"

_pharmaceutics, 2023, doi:10.3390/pharmaceutics15020506_

Round 1

Reviewer 1 Report

This review summarizes recent findings regarding the biological regulatory functions of gut bacterial metabolites. I think that this will be of interest to readers of this journal. 

On the other hand, although "drug delivery systems" is included in the title, the manuscript only mentions an overview of the system, with no new content. Therefore, I think that the manuscript could be improved by adding a section on drug delivery systems, which would be more in line with the title.

Author Response

Response to Reviewer 1 Comments

Point 1: This review summarizes recent findings regarding the biological regulatory functions of gut bacterial metabolites. I think that this will be of interest to readers of this journal.

On the other hand, although "drug delivery systems" is included in the title, the manuscript only mentions an overview of the system, with no new content. Therefore, I think that the manuscript could be improved by adding a section on drug delivery systems, which would be more in line with the title.

Response 1: The drug delivery systems paragraph has been modified, as suggested (highlighted in red).

Reviewer 2 Report

The review on title “Turning microbial AhR agonists into therapeutic agents via drug delivery systems”   lays out a very interesting description of as microbial metabolites play important pathophysiological roles, such as AhR molecular regulation. The manuscript is well performed and easy to follow. I think that this review work adds important findings to consider for AhR regulation and drugs delivery systems. I recommend for acceptance with minor comments addressed below:

1.- Figure 2, I suggest a detailed description in text or in figure caption.

2.- In conclusions, a mention for AhR regulation could be better.

Author Response

Response to Reviewer 2 Comments

The review on title “Turning microbial AhR agonists into therapeutic agents via drug delivery systems”   lays out a very interesting description of as microbial metabolites play important pathophysiological roles, such as AhR molecular regulation. The manuscript is well performed and easy to follow. I think that this review work adds important findings to consider for AhR regulation and drugs delivery systems. I recommend for acceptance with minor comments addressed below:

Point 1.- Figure 2, I suggest a detailed description in text or in figure caption.

Response 1: Thanks for the suggestion. A detailed description of Figure 2 has now been added in the text (in red).

Point 2.- In conclusions, a mention for AhR regulation could be better.

Response 2: Thank you for your suggestion. A mention for AhR regulation has been added in the conclusions (in red).

Round 2

Reviewer 1 Report

The revised manuscript has been reviewed.